# Development of the Activity of Parents of Young Children on Social Networks

**Jana Syrovátková *** **and Antonín Pavlíček** 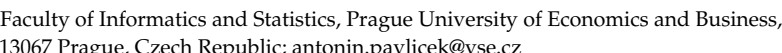

Faculty of Informatics and Statistics, Prague University of Economics and Business,
13067 Prague, Czech Republic; antonin.pavlicek@vse.cz
* Correspondence: jana.syrovatkova@vse.cz

**Abstract:** The paper analyzes the behavior and habits of expectant and new mothers on a specialized pregnancy/parenthood-oriented social network, especially whether and how the pregnancy, and later the age of infants, impact the online activity of mothers. The authors compared almost 5000 parents divided into 23 "term groups"—long-term discussion platforms of parents with the same due month. The age of the child (due date) was taken as the basis for the activity analysis—determining the phases in which the users were more or less active online. Results are shown as charts supported by verification of the following statistical hypotheses: (a) users in later-term groups are less active than those in earlier ones; (b) users' activity peaks around their due dates; (c) users are still very active six months after the due date; (d) activity shortly rises again around the child's first birthday. We concluded that expectant mothers were most active two months before their due dates and around their due dates. After that, the observed activity decreased, with a slight increase around the child's first birthday. Our findings can be useful for sociological and psychological studies, as well as for marketing purposes.

**Keywords:** social network; "birth club" forum; parental activity; parenthood; analysis

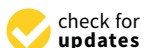



## 1. Introduction

### 1.1. State of the Art

Being a parent is a major responsibility and a highlight of each parent's life. Historically, guidance and advice to new mothers have come from their parents, friends, wider family, and specialized literature. However, new digital technologies are also affecting this aspect of human life profoundly. Nowadays, the Internet and social media are being used to socialize and gather information instead.

Numerous studies of the potential advantages and/or disadvantages of the Internet and social media as sources of parenting information have been published. We used the PRISMA scheme approach in our literature research, and over a hundred potentially interesting records were identified (both through database searching and other sources). We narrowed these down to 56 sources through a screening process, and later further reduced them to 44 full-text articles to be assessed for eligibility. The number of resources included in the synthesis and referred to in our study was 29 [1]. Five articles most related to our research follow. In 2020, the journal *PLOS ONE* published the article "Pregnancy and health in the age of the Internet: A content analysis of online 'birth club' forums" [2], in which the authors analyzed similar Internet platforms as in our study, but instead of statistical analysis/frequency, likes, comments, etc., they focused on the content through natural language-processing methods, and found that the most popular topics were maternal health (45%), baby-related topics (29%), and people/relationships (10%). The *Journal of Medical Internet Research* published the interesting article "Mothers' Perceptions of the Internet and Social Media as Sources of Parenting and Health Information" [3]. The authors of that paper asked soon-to-be mothers questions concerning parenting and health-information sources, searching for reasons why they turn to the Internet and social media for information. It

seemed that the possibility of gathering unrestricted information and a variety of different opinions quickly and anonymously was a key factor, although respondents also appreciated the immediacy of affirmation, support, and tailored information available through social media. It is good to know that mothers did not accept all the information without any doubts, but recognized the need to use reputable sources and double-check information. Another related article, "Using the Internet as a source of information during pregnancy—A descriptive cross-sectional study in Sweden" [4] concluded that in Sweden, the Internet plays a major role for pregnant women in seeking information and getting in touch with like-minded women, since almost all (95%) of the 193 surveyed women used the Internet as a source of pregnancy-related information. Similar findings also were reported in "The Importance of the Internet in Obtaining Health-Related Information in Pregnant Women" [5], in which the authors concluded that, " . . . women use the Internet more often to obtain health-related information, and their tendency towards searching the Internet for information increases during pregnancy." The article "The effect of the Internet on decision-making during pregnancy: a systematic review" [6] found mainly the positives, stating that, "The Internet affects decisions about the type of delivery, drug use in pregnancy, and physical activity. Using the Internet had a positive effect on the decision-making processes of pregnant women, increased their awareness, and had a visible effect on this process." Lastly, the papers "Exploring Women's Health Information Needs During Pregnancy: A Qualitative Study" [7] and "Health information needs, sources of information, and barriers to accessing health information among pregnant women: a systematic review of research" [8] analyzed, in a systematic manner, a majority of articles focused on information needs and sources of information used by women during pregnancy.

The above-mentioned were articles published just in the last two years (2019–2020), but the importance of online resources as an information source was noted even before that, ordered chronologically in [9–21].

In general, it has been proven that the Internet and social media are rapidly becoming important and trusted sources of parental and health information. It even turns out that the use of various information and communication technologies (ICTs) was related to social capital and parenting efficacy among parents of youth [22]. On the other hand, some studies have focused on the relationship between parent media use and child media use and specifically how media may interfere with or strengthen parent–child relationships [21,23,24], while others even looked into racial and ethnic disparities in Internet use for seeking health information among young mothers [25].

Apart from health benefits, online social communities are the ideal place for targeted baby-related marketing. E-shops and producers are in a very good position to address parents, thanks to the quick timing of content publishing [26].

Our paper analyzes online behavior and habits of a specific demographic group—expectant and new mothers, and examines whether and how pregnancy, and later the age of newborns, impact a mother's level of online activity, specifically their behavior on the dedicated pregnancy/parenthood-oriented social network Little Blue Horse (LBH). Preliminary findings based on four "term groups" demonstrated the highest activity around the date of birth [27]. In this article, we have substantially expanded the research to 23 "term groups" and almost 5000 users.

Social network Little Blue Horse (LBH) also was studied in [28], which presented the involvement of paid ambassadors and their influence on social networks aimed at children.

### 1.2. Little Blue Horse (LBH)

Little Blue Horse (LBH) (modrykonik.cz) (accessed on on 22 February 2021) is an extended discussion forum/social-networking site created in 2006. Its main focuses are children, pregnancy, and motherhood-related issues, but also includes women trying to conceive or undergoing artificial insemination. It is similar to some websites such as mumsnet.com or netmums.com [29].

LBH has the traditional characteristics of a typical social network: each registered user has their own profile including a photo and short biography information. The users can publish the latest news, photo albums, and articles on their profiles. Profiles of users who "like" one another are publicly connected, and their content is visible to each other. Public chats, as well as private messages between users, are possible [9].

For discussions, several different formats are used. Simple questions are thematically structured in forums. Users can also follow related threads on the walls of other users. Furthermore, threads can be grouped according to various criteria (problem, experience, birth, etc.). Unlike discussions or walls, groups can be switched to a private mode for members only [30].

A very popular part of LBH is the marketplace, where users buy and sell child-related items. There are usually some competitions for user entertainment, and winning users are given the opportunity to test pregnancy- or baby-related products.

There is also a local wiki system that gathers information related to specific issues within the domain of childcare professionals. This encyclopedia lists frequently asked questions and topics, and its articles are more professional and try to maintain an independent point of view. If the answer cannot be found on the wiki, LBH also contains various professional counseling chat rooms, with the possibility to ask experts (midwives, medical doctors, lactation consultants, gynecologists, dentists) directly.

*1.3. Term Groups*

The first trimester of pregnancy is quite risky (due to miscarriage or fetal development problems). Many pregnant women are therefore reluctant to announce pregnancy publicly this early. Nevertheless, they long to share fears, and gradually joys with someone. Discussions of expectant mothers with a similar due date offer exactly that in the anonymity of the Internet environment. This is the reason why the groups are intended for women connected by a common due date, which in our analysis are labeled by month and year. The LBH assigns the user automatically to such a group upon entering her due date in the system.

There are always individual threads in groups. Responses to threads are listed directly below the relevant thread. It is easy to see the author's username, date, thread, number of likes, and number of comments on the thread.

*1.4. Goals*

The aim of our research was to obtain and compare data from multiple term groups (2 years in total). The main goal was to determine the activity level of individual groups, as well as universal trends.

We also focused on individual users, and examined how many users actually discussed in multiple groups, and how many users discussed over the 2-year period. We also recorded the total number of contributions for each user.

Our paper properly defines and tests the following four hypotheses:

**Hypothesis 1 (H1):** *Users in later-term groups are less active than those in earlier ones;*

**Hypothesis 2 (H2):** *Users' activity peaks around the due date;*

**Hypothesis 3 (H3):** *Users are still very active six months after the due date;*

**Hypothesis 4 (H4):** *Activity shortly rises again around the first birthday.*

## 2. Materials and Methods

Both data collection and data analysis were performed in MS Excel. We focused on 25 "term group" discussions with due dates in 2018 and 2019. Each term group was marked with a month and year according to the due date, i.e., term group I.18 includes users with a due date in January 2018. We used groups with a due date in 2018 and 2019 to ensure that

all the children had already celebrated their first birthday and that all the discussions were active. The October 2019 group was closed, so it was not possible to download the data, therefore we excluded it. Two term groups split, and it was possible to join the data from both subgroups (the January 2018 and March 2019 groups). Therefore, we gathered data for 23 term groups in total.

Using the DownThemAll [31] tool, we downloaded HTML versions of all pages of the relevant discussions (4874 in total). Visual Basic for MS Excel was used to convert the HTML files to a CSV dataset and identify the date of the thread, username, number of comments, and the number of likes, which together with the group identification, were recorded for further analysis.

Pivot tables and contingency graphs were primarily used for the analysis of the obtained data. For statistical analysis, formulas in MS Excel and the statistical tool R were applied [32].

*Analysis of Users*

Table 1 presents the numbers of discussants in each term group. There were 4885 individual users in these 23 groups; 4290 of them were only in one group, while 7 users were in 4 or more groups. A total of 12% of users were in more than one group, while about 3/4 of them were in adjacent groups, and others were in more distant groups—typically, when the mothers had two children shortly apart, they were in two groups accordingly. If we look at how many threads were written by users in their "weaker" groups, i.e., in those where they have fewer posts, it was 4208 posts. If we look at users actually active in more groups—i.e., those who wrote more than 30 posts in this "weaker" group—we found 21 users who wrote 1391 posts, which is only about 1% of all posts. So we can say that most users discussed only in one group connected with their expected due date, and that the number of posts by one user in multiple groups was marginal to the whole.

**Table 1.** Number of individual users in the term groups.

| Term Group (Due Date) | Number of Usernames | Term Group (Due Date) | Number of Usernames |
|:---:|:---:|:---:|:---:|
| I.18 | 348 | I.19 | 174 |
| II.18 | 292 | II.19 | 177 |
| III.18 | 266 | III.19 | 233 |
| IV.18 | 284 | IV.19 | 194 |
| V.18 | 310 | V.19 | 254 |
| VI.18 | 347 | VI.19 | 203 |
| VII.18 | 314 | VII.19 | 199 |
| VIII.18 | 296 | VIII.19 | 220 |
| IX.18 | 284 | IX.19 | 197 |
| X.18 | 234 | | |
| XI.18 | 179 | XI.19 | 181 |
| XII.18 | 163 | XII.19 | 178 |

Table 2 shows how many users had written a specific number of threads. Less than half of the users had more than 5 threads. Thus, it can be said that many users were not in the group for the whole time, but rather arrived randomly during the duration of the group.

**Table 2.** Number of threads written by individual users.

| Number of Threads | Number of Users |
|:---:|:---:|
| 1–5 | 2587 |
| 6–10 | 877 |
| 11–15 | 556 |
| 16–20 | 337 |
| 21–25 | 238 |
| 26–30 | 147 |
| More | 136 |

## 3. Results

The main overview of the data is in Table 3. For each group, we gathered data about all threads, including the number of comments and the number of likes for these threads.

**Table 3.** Global numbers for all groups.

| Term Group | Number of Threads | Number of Likes | Number of Comments | Likes per Thread | Comments per Thread |
|:---:|:---:|:---:|:---:|:---:|:---:|
| I.18 | 26,027 | 283,095 | 229,205 | 10.88 | 8.81 |
| II.18 | 8758 | 54,098 | 73,793 | 6.18 | 8.43 |
| III.18 | 2668 | 15,525 | 25,767 | 5.82 | 9.66 |
| IV.18 | 4879 | 30,314 | 50,033 | 6.21 | 10.25 |
| V.18 | 4917 | 38,306 | 48,660 | 7.79 | 9.90 |
| VI.18 | 8693 | 68,139 | 72,273 | 7.84 | 8.31 |
| VII.18 | 9336 | 79,093 | 80,749 | 8.47 | 8.65 |
| VIII.18 | 8877 | 79,735 | 80,404 | 8.98 | 9.06 |
| IX.18 | 6729 | 58,444 | 58,590 | 8.69 | 8.71 |
| X.18 | 4779 | 42,222 | 43,969 | 8.83 | 9.20 |
| XI.18 | 1513 | 8817 | 13,811 | 5.83 | 9.13 |
| XII.18 | 562 | 3617 | 4030 | 6.44 | 7.17 |
| I.19 | 2570 | 15,328 | 19,495 | 5.96 | 7.59 |
| II.19 | 3036 | 23,618 | 27,240 | 7.78 | 8.97 |
| III.19 | 5174 | 36,924 | 44,081 | 7.14 | 8.52 |
| IV.19 | 2380 | 16,157 | 19,765 | 6.79 | 8.30 |
| V.19 | 3957 | 34,291 | 36,611 | 8.67 | 9.25 |
| VI.19 | 1563 | 12,016 | 15,121 | 7.69 | 9.67 |
| VII.19 | 2786 | 20,291 | 21,545 | 7.28 | 7.73 |
| VIII.19 | 3546 | 24,369 | 28,317 | 6.87 | 7.99 |
| IX.19 | 4178 | 42,762 | 32,680 | 10.24 | 7.82 |
| XI.19 | 2486 | 18,434 | 24,635 | 7.42 | 9.91 |
| XII.19 | 1679 | 11,288 | 15,032 | 6.72 | 8.95 |
| **Total** | **121,093** | **1,016,883** | **1,065,806** | **8.40** | **8.80** |

Table 3 reflects the aggregated data for the entire duration of the group. For comparison, we also calculated the average likes and comments per thread.

The January 2018 group was very active. Of the 2019 groups, the March group was also active, which can be explained by the fact that there were an exceptionally high number of competitions and challenges in this particular group.

There were different numbers of threads in the groups. Understandably, a longer-functioning group had more contributions. Interestingly, the number of comments and likes for individual threads was similar in all groups.

Looking at the data, the groups with a due date in 2019 seemed to be less active than the groups with a due date in 2018. By activity, we mean the number of threads, comments, and likes—all these parameters were lower. Why this occurred would be an interesting topic for further research. We tested hypothesis H1 in the following forms: The mean of number of threads in the second year was the SAME as the first-year number of threads

against the hypothesis that the mean of the second year was DIFFERENT from the first year, with comments and likes analogically.

For testing, we used a two-sample *t*-test to compare distributions with unequal amounts of data. We calculated variances and means in Excel. In R, we used an F-test to verify that there was unequal variance. Than we used a *t*-test for unequal variances, showing in Table 4.

**Table 4.** T-statistics for the comparison of the results for the 2018 and 2019 groups.

| Term Groups | Statistical Indicator | Number of Threads | Number of Like | Number of Comments |
|---|---|---|---|---|
| 2018 | Mean | 178 | 1540 | 1588 |
| | Variance | 21,463 | 2,683,829 | 1,624,493 |
| | Number of data | 12 | 12 | 12 |
| 2019 | Mean | 109 | 842 | 934 |
| | Variance | 1421 | 153,496 | 96,508 |
| | Number of data | 11 | 11 | 11 |
| Comparison | F-statistics | 15.1 | 17.5 | 16.8 |
| | Welch t-stat | 1.58 | 1.43 | 1.72 |
| Rejection | | No | No | No |

T-statistics displayed less than a critical value, and the *p*-value was greater than 0.05, so it can be seen that although the groups were less active from the point of view of the second year, probably due to the high variance, the hypothesis of identical activity cannot be rejected on a significance level of 0.05.

### 3.1. Analysis of Threads

The main purpose of the analysis was to examine the activity of users in the relevant months before, around, and after childbirth. We used charts for the basic data overview. Charts of the number of threads, the number of comments, and the number of likes in the respective months were used for this purpose.

Figure 1 shows the number of threads in each term group for each month, calculated according to the due date. Figures 2 and 3 also show the individual term groups for each month and the number of likes and the number of comments on the threads. For better data visualization, we combined some groups with a similar course and always took the average of these groups. Term groups with a due date in 2019 are represented by a dashed line.

In each of the groups, the first contributions were 8 to 9 months before the respective due date. This means that women typically started discussing at the early stage of pregnancy. In the Czech Republic, a woman takes maternity leave 6–8 weeks prior the due date. The charts show a sharp increase in user activity about 2 months before the due date, which means there was a strong connection with taking maternity leave. Maximum activity was at the moment of the birth of a child. This was followed by a gradual decline. The less-significant decrease was in the number of comments. For contributions and likes, the decline was sharp at first and then slow until about 11 months of age. In the child's first year, there was a slight increase, but then a decrease again.

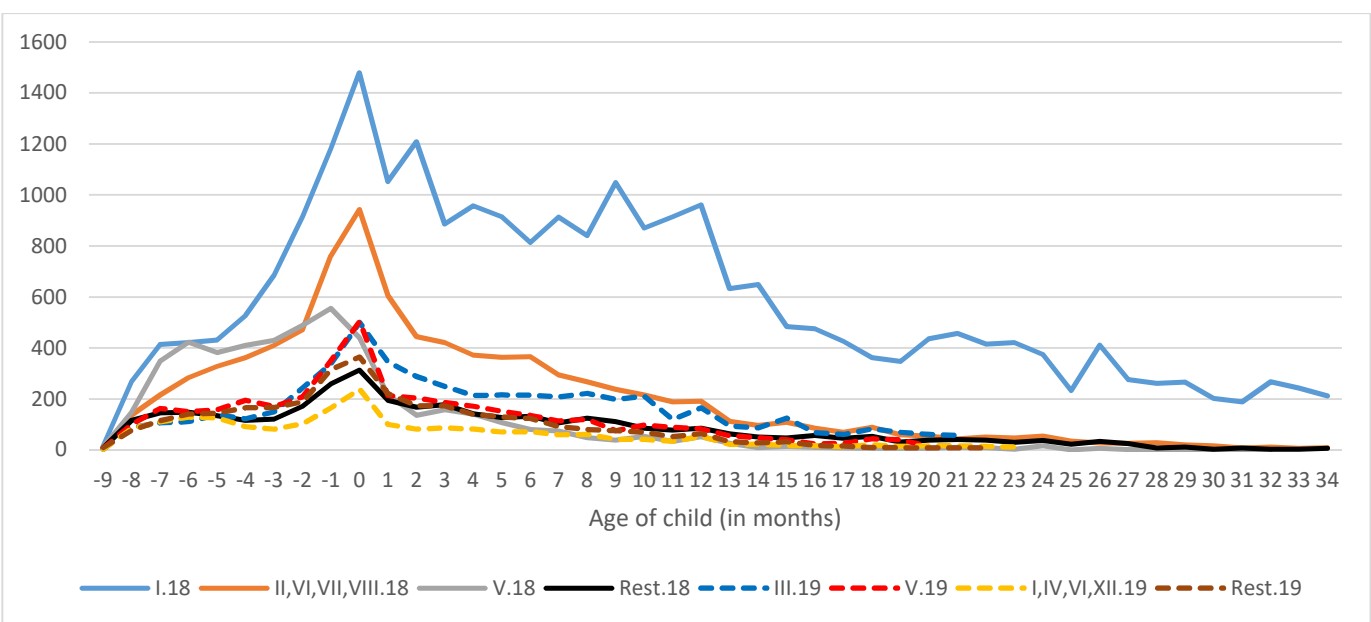

**Figure 1.** A chart showing the number of threads per month in groups from the beginning of the group. For some groups, the average was taken.

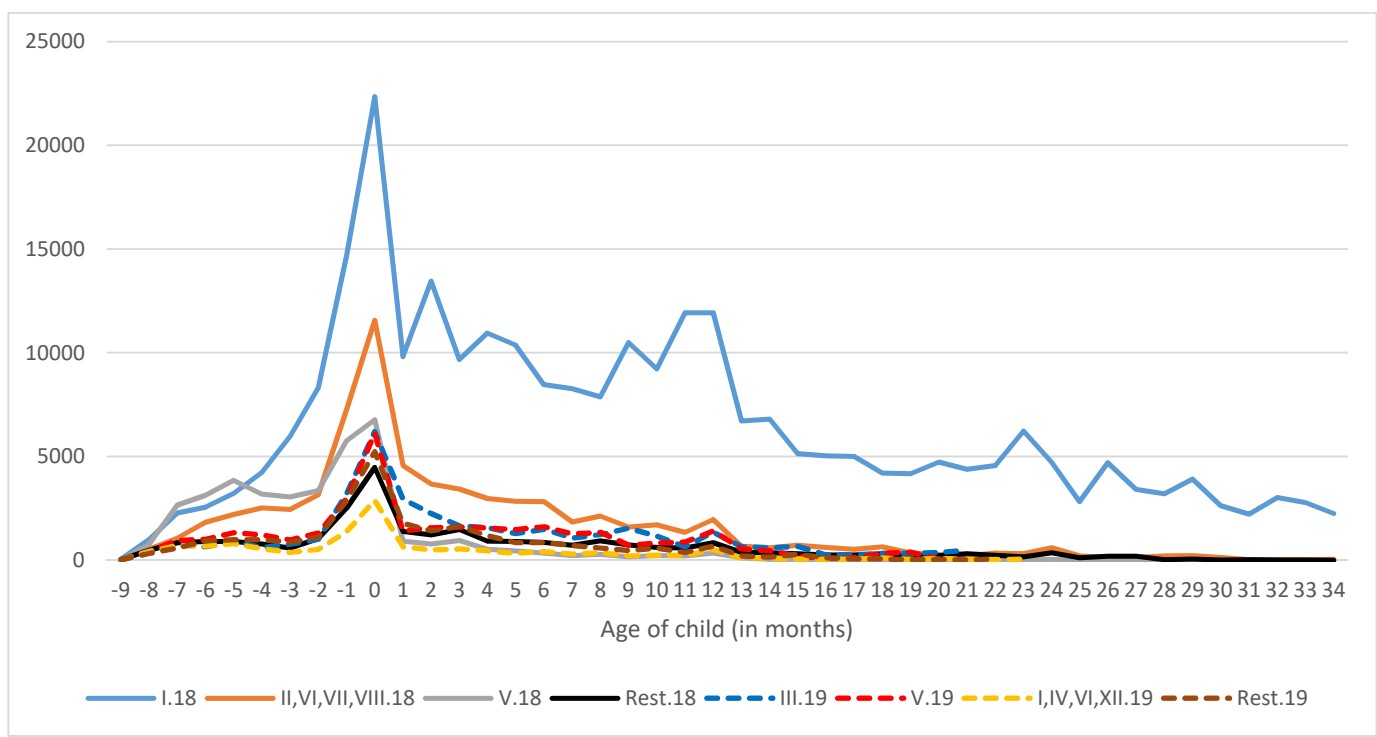

**Figure 2.** A chart showing the number of likes to threads per month in groups from the beginning of the group. For some groups, the average was taken.

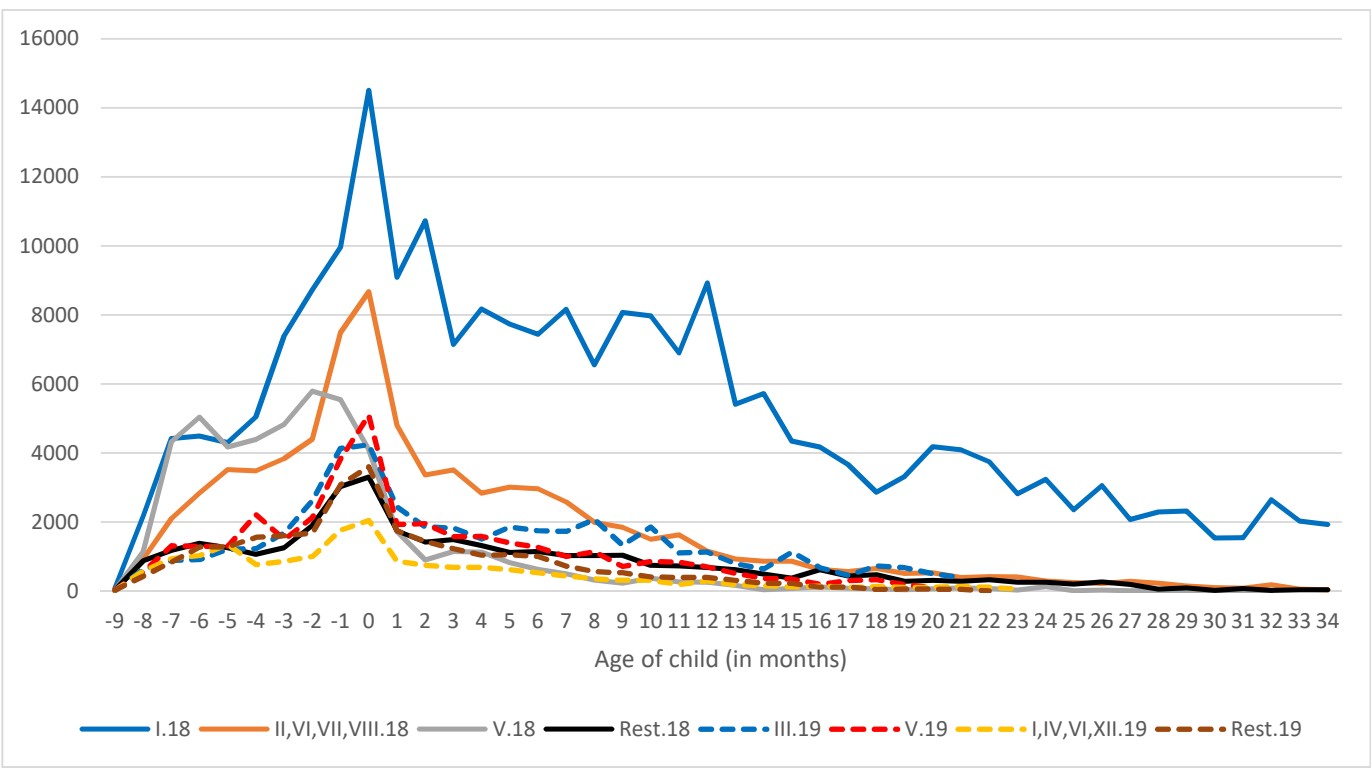

**Figure 3.** A chart showing the number of comments to threads per month in groups from the beginning of the group. For some groups, the average was taken.

### 3.2. Activity Hypothesis Testing

Now we are going to look at statistical hypotheses, which we declared very generally in the Introduction. Here we define them exactly in mathematical terms.

We will take these exact forms of the hypotheses:

**Hypothesis 2$_1$ (H2$_1$):** *Average number of threads 2 months before the due date is the same as the number of threads in the month of the due date.*

**Hypothesis 2$_2$ (H2$_2$):** *Average number of threads 4–6 months after the due date is the same as the number of threads in the month of the due date.*

**Hypothesis 3 (H3):** *Average number of threads 4–6 months after the due date is the same as the number of threads 9–11 months after the due date.*

**Hypothesis 4 (H4):** *Average number of threads 9–11 months after the due date is the same as the number of threads 12 months after the due date.*

We will test everything against the hypothesis HA: An average number of threads is different.

We use a paired *t*-test to determine whether the difference was equal to zero at a significance level 0.05. We used R software and Excel. The results are shown in Table 5.

**Table 5.** Statistical values of paired *t*-test for differences.

|  | Mean of Differences | Variance of Differences | t Statistics | Rejections on a Significance Level 0.05 |
|---|---|---|---|---|
| **H2$_1$** | −159 | 17,882 | −5.71 | Yes |
| **H2$_2$** | 294 | 30,940 | 8.01 | Yes |
| **H3** | 65 | 3374 | 5.38 | Yes |
| **H4** | 1 | 377 | 0.26 | No |

We can see that the activity of groups was highest around the due date, then gradually decreased. We cannot reject (on a significance level 0.05) the hypothesis that the birthday was comparable to the activity in previous months, and not that it was growing, as it would appear from the charts.

## 4. Discussion and Conclusions

In contrast to a previous study [27], which examined only 4 term groups from one year, this extended study examined 23 groups covering almost the entire period of 2 years. The following characteristic behavior in all groups was confirmed:

- The groups begin to be active as soon as the pregnancy was detected, with an increase in activity 2 months before delivery.
- There was a sharp and statistically significant increase in activity around childbirth, followed by statistically significant inactivity.
- It appeared that the activity then increased in the month of first birthdays, but this increase was not statistically significant.

Regarding the analysis of users, most users were primarily passive (lurking) and with quite a low level of active contributions. This can be explained by the fact that mothers visit social media irregularly, primarily to get a piece of advice with an urgent problem, and are not active for the entire duration of the group.

Our findings and results confirmed previous studies [2,4,6] that showed the majority of pregnant women used online media to seek information during pregnancy, around their due dates, as well as postpartum. Other researchers so far have used primarily self-reporting measures to see how and why women turn to an online environment, while we analyzed quantifiable online content that women actually generated. However, we reached a similar conclusion: The Internet and social media have become frequent and trusted sources of pregnancy and parenting information that women turn to ever more often, especially around the time of birth.

**Author Contributions:** Conceptualization, A.P. and J.S.; methodology, J.S. and A.P.; formal analysis, A.P.; investigation, A.P. and J.S.; resources, A.P.; programming parser, J.S.; data curation, J.S.; writing—original draft preparation, J.S.; writing—review and editing, A.P.; visualization, J.S.; supervision, A.P.; project administration, A.P.; funding acquisition, J.S. and A.P. All authors have read and agreed to the published version of the manuscript.

**Funding:** This paper was processed with a contribution from the University of Economics in Prague, IG Agency, OP VVV IGA/A, CZ.02.2.69/0.0/0.0/19_073/0016936, grant number F4/05/2021.

**Institutional Review Board Statement:** Not applicable.

**Informed Consent Statement:** Not applicable.

**Data Availability Statement:** The dataset used in this article is available at https://drive.google.com/file/d/1IBFsp4Js9EbIZDm9fpiJ7vzjnGQOCsI5/view?usp=sharing. (accessed on 22 February 2021) The program codes with parser are available at https://drive.google.com/file/d/1PvOszR-C9gJPWUOUqD51ewSANirbE79V/view?usp=sharing. (accessed on 22 February 2021).

**Conflicts of Interest:** The authors declare no conflict of interest.

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
