# Peer review of "Development of the Activity of Parents of Young Children on Social Networks"

_information, doi:10.3390/info12030102_

Round 1

Reviewer 1 Report

The research article, presented by the authors, uses an approach that, due to its originality, may be of interest to the journal's reader.

The manuscript presents a suitable organization and structure. And although, its reading is difficult at some stages, it is in general terms adequate. The work stands out for its methodological rigor, explaining the techniques used and providing sufficient evidence to contrast the postulated hypotheses.

Regarding the points of improvement, in author's opinion the following issues should be addressed.

Introduction.

The estate of art should be more solid. A estate of art that provides only eight bibliographic references cannot be considered optimal, in any case.

Results.

The evidence provided is adequate and the way to do it also. However, tables and figures have formatting problems. The tables do not follow homogeneous formats, even using different types of fonts in some cases. Regarding figures, it is recommended to adjust their format and features to improve the visualization and facilitate the compression of the information shown in them.

Discussion.

The “Discussion” section is not such discussion. An academic discussion involves a dialogue comparing the results obtained in the study with the findings of previous research. This fact does not occur. It is recommended to reinforce this section by comparing the conclusions of the present work with those of the studies of other authors. That implies, obviously, providing the new references required for this purpose.

Marketing purposes addressed in the abstract.

Even though the abstract presents as a contribution of the work the applicability of its results to the marketing field. This point is poorly developed in the manuscript (only two brief hints in lines 50 and 247). In this sense, the author's recommendation is to either develop this point further or eliminate it from the manuscript.

Author Response

Dear Reviewer,

thank you for your valuable comments and suggestions, we have tried to addressed them as much as possible. Comments on individual points are attached in the document. 

Best regards

A.P. and J.S.

Reviewer 2 Report

Introduction is good in that it grabs the reader's interest, but some work is needed.

Line 30-32 could be removed and it would strengthen the paragraph and the point that is being made.

Reference 7

Syrovátková, J.; Novák, R. Parental Activity on Social Networks.; 2020; pp. 231–236.

I could not find this article online, it seems like a conference proceeding, I wanted to reference it as it seemed pertinent to the current work.

Line 96, I believe it should be username instead of nickname in this case:

nickname is a familiar, invented given name for a person or thing used instead of the actual name of the person or thing while username is (computing) a person's identification on an individual computer system.

https://wikidiff.com/nickname/username#:~:text=As%20nouns%20the%20difference%20between,on%20an%20individual%20computer%20system.

Line 99-100 Does not read properly. It would flow better

“The aim of our research was to obtain and compare data from multiple term groups.”  “compare them in between” doesn’t work grammatically.

Line 100:

The main goal is to find out how active the individual groups are during their lifetime, and which unifying trends can be traced within all groups.

That includes a comma splice.

Would read better as:

The main goal is to find out the activity level of individual groups as well as universal trends.

Line 102:

“The long-term activity of the group is also an interesting figure.”  This is vague and unclear.  Not sure what group you are referencing and what is meant by an interesting figure.

Line 104 Need to be more specific on what is meant by “in the long run”

Line 119 should provide some reference for DownThemAll

Line 121 should be username

Section 2.1 provides good background.

Lines 158-167

You shouldn’t have to provide code like this in the paper.  If requested put it in a supplemental table.  Just report the statistics.  However there is an error in the code:

You use

t.test(year2018,year2019, var.equal=TRUE)  But you state that you would use

“a two-sample t-test to compare distributions with unequal variances and the amount of data.”

Line 200

Should be Introduction rather than Chapter 1

Author Response

(The authors gave the same response as above.)

Reviewer 3 Report

This manuscript proposes good research questions and new avenues of research: this is its main advantage/contribution.

However, it can be improved, mainly when it comes to justify why some measures/methodological approches have been selected, if compared with other approaches.

In fact, if on the one side it is acceptable that author(s) have chosen a certain methodology and the corresponding measures, on the other side, this choice needs more scientific support form extising published works.

For example, focus more on the contributions on centrality measures (and other measures: iI provide you with som references you may wish to add to your work. The key point is discussing them, and explaining why they have not been selected:

  • Aramo-Immonen, H., Jussila, J., Huhtamäki, J., Visualizing Informal learning behavior from conference participants twitter data,(2014) ACM International Conference Proceeding Series, pp. 603-610, doi: 10.1145/2669711.2669962
  • Bastian, M., Heymann, S., Jacomy, M., Gephi: An Open Source Software for Exploring and Manipulating Networks (2009), Proceedings of the Third International ICWSM Conference.
  • Brandes, U., A faster algorithm for betweenness centrality (Open Access)(2001), Journal of Mathematical Sociology, 25 (2), pp. 163-177, doi: 10.1080/0022250X.2001.9990249
  • Jussila, J., Huhtamäki, J., Kärkkäinen, H., Still, K. Information visualization of Twitter data for co-organizing conferences, (2013) Proceedings of the 17th International Academic MindTrek Conference: "Making Sense of Converging Media", MindTrek 2013, pp. 139-145, doi: 10.1145/2523429.2523482
  • Palmieri, R., Giglio, C., Turkish Online Journal of Educational Technology, Volume 2015, 1 July 2015, Pages 279-287, Informal learning in online social network environments: An evidence from an academic community on facebook
  • Palmieri, R., Giglio, C., Turkish Online Journal of Educational Technology, Volume 2015, 2015, Pages 698-703, Using social network analysis for a comparison of informal learning in three Asian-American student conferences

Author Response

(The authors gave the same response as above.)

Reviewer 4 Report

  • Line 106, These hypotheses are too general. How did you define the concepts of active, around, later and earlier? The authors need to define the response variable and the factors. Then state these hypotheses in term of these variables.
  • Line 117, “Therefore we have data for 23 term groups in total”. The authors should add a table like Table 1 but with 2 columns Term group and Due Date for clarity.
  • Line 130, 131: “7 users are in 4 or more groups. 12% of users are in more than one group, about 3/4 of them are in adjacent groups, others are in more distant groups”. Does this overlapping affect the result? One of the assumptions of t-test is independence observation.
  • Line 151, “Data show, that the second year is always less active than the first one.” How did the authors define active? Which part of data support that statement?
  • Line 153, “the mean of the second year…”. Which variable did the authors refer to? Please see the above comment about the set of hypotheses
  • Line 155, “For testing, we will use a two-sample t-test to compare distributions with unequal variances and the amount of data”. The code didn't match with the discussion. The authors used t-test for equal variance (by setting var.equal = TRUE). Did the authors check and validate the assumptions of t-test?
  • The R code should be put in appendix and save the space for discussion.
  • Line 172, incorrect terminology “confidence level”. It should be significance level.
  • Line 182,183: “We highlighted 3 special groups – March and April 2018 and March 2019. These groups are a special different activity than the groups around them”. It’s impossible to see the highlighted part in the charts. The authors should describe how different they are.
  • Line 200, “which we declared in Chapter 1”. What chapter 1 referred to?
  • Line 215, the authors should check and validate the assumption of paired t-test.

Author Response

(The authors gave the same response as above.)

Round 2

Reviewer 1 Report

I reviewer´s opinion the manuscript has been significantly improved and all the issues addressed during the first round have been covered.

Reviewer 2 Report

The paper has gone through a marked improvement since the first version.  There were still a few grammatical issues, but not enough to distract from the paper's content.  The one minor correction I would suggest is that you refer to R as 

 "In the R" around line 198.  It should be "In R," or in that case, "We used F-tests in R to verify ..."

Reviewer 3 Report

Authors' response are reasonable: accept.